# Habitual Dietary Intake Affects the Altered Pattern of Gut Microbiome by Acarbose in Patients with Type 2 Diabetes

**DOI:** 10.3390/nu13062107

**Published:** 2021-06-19

**Authors:** Fumie Takewaki, Hanako Nakajima, Daiki Takewaki, Yoshitaka Hashimoto, Saori Majima, Hiroshi Okada, Takafumi Senmaru, Emi Ushigome, Masahide Hamaguchi, Masahiro Yamazaki, Yoshiki Tanaka, Shunji Nakajima, Hiroshi Ohno, Michiaki Fukui

**Affiliations:** 1Department of Endocrinology and Metabolism, Graduate School of Medical Science, Kyoto Prefectural University of Medicine, 465, Kajii-cho, Kawaramachi-Hirokoji, Kamigyo-ku, Kyoto 602-8566, Japan; fumi12112000@yahoo.co.jp (F.T.); tabahana@koto.kpu-m.ac.jp (H.N.); saori-m@koto.kpu-m.ac.jp (S.M.); conti@koto.kpu-m.ac.jp (H.O.); semmarut@koto.kpu-m.ac.jp (T.S.); emis@koto.kpu-m.ac.jp (E.U.); mhama@koto.kpu-m.ac.jp (M.H.); masahiro@koto.kpu-m.ac.jp (M.Y.); michiaki@koto.kpu-m.ac.jp (M.F.); 2Department of Neurology, Graduate School of Medical Science, Kyoto Prefectural University of Medicine, 465, Kajii-cho, Kawaramachi-Hirokoji, Kamigyo-ku, Kyoto 602-8566, Japan; tkwk0910@koto.kpu-m.ac.jp; 3Department of Diabetes and Endocrinology, Matsushita Memorial Hospital, 5-55 Sotojima-cho, Moriguchi 570-8540, Japan; 4R&D Center, Biofermin Pharmaceutical Co., Ltd., Kobe 650-0021, Japan; tanaka_yoshiki@biofermin.co.jp (Y.T.); nakajima_shunji@biofermin.co.jp (S.N.); ohno_hiroshi@biofermin.co.jp (H.O.)

**Keywords:** type 2 diabetes, gut microbiome, α-Glucosidase inhibitors, diet, dietary habit

## Abstract

The aim of this research was to reveal the characteristics of gut microbiome altered by acarbose intervention in Japanese patients with type 2 diabetes (T2D) and its possible association with habitual dietary intake. Eighteen patients with T2D were administered acarbose for four weeks. The abundances of two major phyla, namely Actinobacteria and Bacteroidetes, were reciprocally changed accompanied by the acarbose intervention. There were also significant changes in the abundances of ten genera, including the greater abundance of *Bifidobacterium*, *Eubacterium*, and *Lactobacillus* and the lower abundance of *Bacteroides* in the group after the intervention than that before the intervention. Hierarchical clustering of habitual dietary intake was performed based on the pattern of changes in the gut microbiota and were classified into distinct three clusters. Cluster I consisted of sucrose, cluster II mainly included fat intake, and cluster III mainly included carbohydrate intake. Moreover, the amount of change in *Faecalibacterium* was positively correlated with the intake of rice, but negatively correlated with the intake of bread. The intake of potato was negatively correlated with the amount of change in *Akkermansia* and *Subdoligranulum*. Acarbose altered the composition of gut microbiome in Japanese patients with T2D, which might be linked to the habitual dietary intake.

## 1. Introduction

The number of patients with type 2 diabetes (T2D) is increasing in developed countries and exceeds 10 million in Japan. Since genetic risk factors for T2D have mostly remained the same in the Japanese population, environmental risk factors are thought to play significant roles. Recently, several studies have revealed that the modern lifestyle appears to influence the gut microbiome, which may have an important role in the pathogenesis of T2D [1,2,3]. Accumulating evidence suggests that anti-diabetic drugs have significant effects on the gut microbiome and modify the T2D pathogenesis [4,5,6,7]. In noninsulin-dependent T2D, α-Glucosidase inhibitors (α-GI) are commonly used drugs [8,9]. Moreover, α-GI inhibits carbohydrate hydrolysis in the upper small intestine by binding to maltase-glucoamylase and sucrase-isomaltase, and consequently delays and reduces the absorption of glucose into the bloodstream. Unabsorbed carbohydrates then flow into the large intestine and possibly change the structure and function of the gut microbiome because microbial species living in the gut convert the wide range of carbohydrates into simple sugars and use them for energy [10,11].

Previous studies that addressed the possible changes of the gut microbiome in patients with T2D associated with acarbose intervention have revealed the specific microbial species that might influence this disease [7,12,13,14]. Gu et al. reported that acarbose increased the abundance of *Bifidobacterium* and *Lactobacillus*, whereas the abundance of *Bacteroides* was decreased at the genus level; as a result, the abundance of microbial genes, which are associated with the bile acid metabolism, and plasma bile acid composition are altered. The association between the hypoglycemic effects of acarbose and bile acid signals was studied using genetically modified murine models [15]. Zhang et al. also reported that acarbose contributed to the significant increase of *Bifidobacterium* and *Lactobacillus* at the genus level. However, it remains unclear about the potential role of α-GI on the gut microbiome of patients with T2D in the Japanese population and whether those microbial changes are associated with the habitual dietary intake.

In the present study, we analyzed the alterations in the gut microbiome associated with the intervention of acarbose in Japanese patients with T2D. We also performed analysis to correlate these microbial changes with their habitual dietary intake. The results revealed some alterations in the gut microbiome associated with acarbose intervention and suggested the potential role of dietary habit on them.

## 2. Materials and Methods

### 2.1. Study Population

Eighteen patients with T2D (mean age: 64.5 (48–75)) were enrolled randomly from October 2018 to July 2019 in Kyoto Prefectural University of Medicine (KPUM) (Kyoto, Japan). T2D was diagnosed by using the Report of the Expert Committee on the Diagnosis and Classification of Diabetes Mellitus [16]. The participants were administered 150 or 300 mg/day of acarbose for four weeks. All participants were naive to acarbose therapy and administered 50 mg three times per day (n = 13) or 100 mg three times per day (n = 5). The dose was determined by the attending doctors considering the side effects. Other drugs were unchanged during the research period. Fecal and blood samples were gathered a day before and four weeks after the start of intervention. The patients with estimated glomerular filtration rate (eGFR) of <60 mL/min/1.73 m^2^; patients whose oral hypoglycemic medications changed within 3 months prior to the participation, patients consumed glucagon-like peptide 1 agonist or antibiotics within 3 months prior to the participation; and patients who had heart failure, liver failure, and gastrointestinal diseases (ulcerative colitis, short bowel syndrome, Crohn disease, celiac disease, or diverticulosis) were not included. Participants were instructed not to change their eating habits or lifestyle during the study.

### 2.2. Ethical Consideration

This study was approved by the ethics committee of KPUM. (Approval number ERB-C-1166-2) and was conducted in accordance with the principles of the Declaration of Helsinki. Signed informed consent was obtained from all subjects who provided specimens.

### 2.3. Data Collection and Variables

Age, sex, body mass index, and blood pressure (both systolic and diastolic) data were collected from all the participants. Data are represented as median (range). The mean disease duration was 10.6 ± 6.5 years. One subject received abdominal surgery. The ratio of never smokers to past smokers to current smokers was 11:6:1. The ratio of never drinkers to opportunistic drinkers to habitual drinkers was 8:8:2. Retinopathy was evaluated as follows: no- (NDR), simple- (SDR), pre-proliferative- (PPDR), or proliferative diabetic retinopathy (PDR) [17]. The ratio of NDR to SDR to PPDR to PDR was 15:1:1:1. Nephropathy was graded as follows: stage 1, 2, and 3. This grading was based on the urinary albumin excretion, which was the average of three urine collections [18]. The ratio of stage 1 to stage 2 to stage 3 was 14:3:1. Neuropathy was evaluated by using the diagnostic criteria for diabetic neuropathy proposed by the Diagnostic Neuropathy Study Group [19]. Three patients had neuropathy. No patient had a past history of cerebrovascular disease or coronary artery disease. Blood samples for analyses, including analysis of hemoglobin A1c, creatine, aspartate transaminase, alanine aminotransferase, glutamyltransferase, total-cholesterol, triglycerides, HDL-cholesterol, LDL-cholesterol, and white blood cell count, were obtained from the participants and are shown in Table 1. The Japanese Society of Nephrology equation was used for evaluating GFR: eGFR = 194′ Creatine–1.094′ age–0.287 (mL/min/1.73 m^2^) (′ 0.739, if patient is female) [20]. The patients were surveyed with regarding the medication for diabetes, dyslipidemia, hypertension, histamine H2 blockers, and proton pump inhibitors. Among the 18 participants, 15 received dipeptidyl peptidase-4 inhibitors, 14 received metformin, 6 received sulfonylurea, 2 received glinide, 1 received thiazolidine, 10 received sodium-glucose transporter 2 inhibitors, 2 received insulin, 7 received angiotensin converting enzyme inhibitors or angiotensin II receptor blockers, 3 received statin, and 2 received proton pump inhibitors. No patient received histamine H2-receptor blockers. The data on the habitual dietary intake were obtained by a brief-type self-administered diet history questionnaire (BDHQ) before acarbose administration. The details of BDHQ have been explained previously [21]. Briefly, it evaluates the dietary habits in the past month in which the food and beverage items listed are all common in Japan [21]. In this study, we evaluated essential habitual dietary intake, such as carbohydrates, animal and vegetable proteins, animal and vegetable fats, saturated fats, polyunsaturated and monounsaturated fats, soluble and insoluble dietary fibers, and sucrose. Furthermore, staple foods intake, namely rice, bread, noodle, and potato, was also evaluated.

### 2.4. Bacterial DNA Extraction from Feces

Each patient collected a stool sample at home for the present study, using a feces collection kit (Techno Suruga Lab, Shizuoka, Japan) that was pre-filled with 5 mL of stool DNA stabilizer. Samples were immediately stored at household refrigerator and then delivered to the Kyoto Prefectural University of Medicine. The fecal samples were stored at 4 °C until DNA extraction. Extraction of bacterial DNA was performed as described previously [22]. Twenty milligrams of frozen sample was suspended in sterile phosphate-buffered saline solution and washed 3 times. Bacterial pellet was suspended in 450 µL of extraction buffer (100 mM Tris-HCl, 40 mM EDTA, pH 9.0). Subsequently, 500 μL of buffer-saturated phenol and 50 µL of 10% sodium dodecyl sulfate with 300 mg of 0.1 mm diameter glass bead were added. Lysis step, shaking (4000 rpm for 10 s) using Micro smash MS-100 (TOMY, Tokyo, Japan) followed by incubation at 65 °C for 10 min, was conducted twice. The lysate was centrifuged at 14,000× *g* for 5 min, and 400 µL of supernatant was added to equal amount of phenol to chloroform to isoamylalcohol (25 to 24 to 1). After gently mixing, centrifugation was performed at 20,000× *g* for 10 min. Bacterial DNA was precipitated by 25 µL of 3 M of sodium acetate (pH 5.2) with 250 µL of ice-cold isopropanol and pelleted by centrifugation at 20,000× *g* for 15 min. The DNA pellet was washed by 70% ethanol, dried, and then dissolved with 1 mL of TE (10 mM Tris-HCl, 1 mM EDTA, pH 8.0).

### 2.5. DNA Sequence Analysis

The microbial community structure in feces was determined by 16S rDNA analysis, using MiSeq (Illumina, San Diego, CA, USA). The amplification of V3-V4 region of 16S rDNA was performed by forward primer (5′-TCGTCGGCAGCGTCAGATGTGTATAAGAGACAGCCTACGGGNGGCWGCAG -3′) and reverse primer (5′-GTCTCGTGGGCTCGGAGATGTGTATAAGAGACAGGACTACHVGGGTATCTAATCC -3′), which were bound with protrusion Illumina adapter consensus sequences. The initial PCR reaction program was followed: 95 °C for 3 min, followed by 25 cycles consisting of 95 °C for 30 s, 55 °C for 30 s, and 72 °C for 30 s. The reaction was completed with a final extension of 5 min at 72 °C on a Veriti thermal cycler (Thermo Fisher Scientific, Waltham, MA, USA) after 25 cycles. Purification of amplicon was performed by AMPure XP magnetic beads (Beckman Coulter, Brea, CA, USA). Multiplexing was then performed by The Illumina Nextera XT Index kit (Illumina) with dual 8-base indices. To integrate two unique indicators to the 16S amplicons, PCR reactions were performed. Cycling conditions were 95 °C × 3 min, 95 °C × 30 s, 55 °C × 30 s, and 72 °C × 30 s for 8 cycles, and finally an extended cycle of 72 °C × 5 min. After purification with AMPure XP beads, purified barcoded library was fluorometrically quantified by using a QuantIT PicoGreen ds DNA Assay Kit (Invitrogen, Paisley, UK). Next, 10 mM Tris-HCl (pH 8.0) was loaded to libraries and then diluted to 4 nM; then we preserved the same volume for multiplex sequencing. To improve base calling during sequencing, the multiplexed library pool (10 pM) was spiked with 40% PhiX control DNA (10 pM). Sequencing was performed by a 2 × 250-bp paired-end run on MiSeq platform with MiSeq Reagent Kit v2 chemistry (Illumina). QIIME was used for the elimination of chimera and low-quality sequences, construction of operational taxonomic units (OTU number), and taxonomy assignment [23]. In brief, 50,000 raw reads were randomly gathered from the sequence files for each sample and combined by using fastq-join with the default settings. The paired-end data were merged by fastq-join, and only sequences with a Phred score of <25 were quality filtered, and then the chimera read was detected by using usearch61. Both were performed with the default settings. We randomly selected 5000 reads per sample and analyzed them in order to minimize the overestimation of the species richness in the clustering due to intrinsic sequencing error. The Good’s coverage index of the 5000 reads per sample in this study exceeded 0.98, indicating a high coverage degree which was sufficient reads number for this fecal microbiome analysis [24]. OTU numbers were constructed by clustering with a 97% identity threshold. Using UCLUST with a ≥97% identity, we designated the representative reads for each OTU to the database of 16S rRNA gene. Comparison of each taxon in the gut microbiome was carried out at the genus-level. Beta diversity was evaluated by computing the Weighted and Unweighted UniFrac distance between samples [25]. To compare the differences in the overall bacterial gut microbiome structure, reduction of the dimension of the distance matrix was performed by PCoA. The alpha diversity was evaluated by using the Shannon index, observed OTU number, chao1, and ACE.

### 2.6. Statistical Analysis

Statistical analyses were performed by R version 3.1.3.25, Prism software (GraphPad Software, San Diego, CA, USA), or Excel version 16.35. Paired *t*-test was performed to compare the microbiome data between the groups. Spearman’s correlation coefficient analysis was used to evaluate correlations. Mann–Whitney U test was used for the comparison of patient demographic data. A PERMANOVA was used for comparing the overall microbiome structure between the groups. Data are shown as mean ± SEM. The significance level was set at *p*-value < 0.05.

## 3. Results

### 3.1. Demographic Profiles of Subjects

Eighteen patients with T2D were recruited for this study and all patients completed the intervention. Among them, two experienced acid reflux, seven experienced indigestion, three experienced diarrhea, and four experienced constipation. The demographics of the subjects before and four weeks after the intervention of acarbose are shown in Table 1.

### 3.2. Comparison of Alpha and Beta Diversities in the Gut Microbiome before and after the Intervention of Acarbose

We obtained high-quality 16S reads from the eighteen subjects. We then randomly selected and used 5000 reads per sample, accounting for a total of 90,000 reads from 18 samples, for evaluation of the alpha and beta diversities (see Section 2.5). The four indices (observed OTU number, Chao-1, ACE, and Shannon index) did not show significant differences between before and after the intervention of acarbose (Appendix A).

We next evaluated the beta diversity by using the UniFrac distance and the 16S data. PERMANOVA did not show any significant differences in the overall gut microbiome structure (beta-diversity) before and after the intervention based on both weighted and unweighted UniFrac distances (Appendix A).

### 3.3. Assignment and Identification of Microbial Species Significantly Associated with Acarbose Intervention

We analyzed and compared the microbial abundance at various taxonomic levels based on the 16S data. Taxonomic assignment and abundance quantification in each sample were performed by mapping the 16S reads to the microbial 16S and genome databases. The phylum-level assignment identified six phyla with an average relative abundance of ≥0.1% in both groups before and after the intervention, accounting for 98% of the total abundance (Appendix A). There were significant changes in the abundance of Bacteroidetes and Actinobacteria between the before the intervention and after the intervention groups (Figure 1). The genus-level assignment identified 32 genera with an average relative abundance of ≥0.5% in both groups, accounting for 90% of the total abundance (Appendix A). Among them, the abundances of *Bifidobacterium*, *Eubacterium*, *Megasphaera*, and *Lactobacillus* were significantly higher in the group after the intervention than in the group before the intervention (Figure 2 and Appendix A), whereas the abundances of *Bacteroides*, *Blautia*, *Prevotella*, *Clostridium, Phascolarctobacterium*, and *Lachnoclostridium* were significantly lower in the group after the intervention than in the group before the intervention (Figure 2 and Appendix A). The abundances of *Akkermansia* did not show significant differences between before and after the intervention of acarbose (Appendix A).

### 3.4. Association between Habitual Dietary Intake and Changes in Microbial Data Related to Acarbose Intervention

We also examined Spearman’s correlation between the habitual dietary nutrient intake and changes in major microbial data (X delta; X after intervention − X before intervention) including the observed OTU number, the Shannon index, and the abundance of the top four phyla and the top five genera associated with the intervention of acarbose in all the 18 patients with T2D. In Figure 3, each Spearman’s r, which reflects the degree of positive and negative correlation between changes in the composition of the gut microbiome and habitual dietary nutrient intake, was converted to z-score in order to apply to hierarchical clustering. The data of the dietary intake values are shown in Appendix A. The row z-score values are shown in Appendix A. The 12 dietary nutrient items were used for hierarchical clustering based on the pattern of the alterations of the major gut microbial data, and subsequently classified into three clusters. Cluster I consisted of sucrose, cluster II included animal protein, animal fat, saturated fat, cholesterol, monounsaturated fat, polyunsaturated fat, and vegetable fat, and cluster III included vegetable protein, insoluble dietary fiber, soluble dietary fiber, and carbohydrates, suggesting that dietary habits on sucrose, lipid, and carbohydrate influenced in a different way on the changes in gut microbiome associated with acarbose.

Next, we examined the Spearman’s correlation between the intake of various staple foods and the amount of changes in the abundance of 32 genera with an average relative abundance of ≥0.1%, associated with the intervention of acarbose (Figure 4 and Table 2). The intake of rice was positively correlated with the relative abundance of *Faecalibacterium* delta (Figure 4 and Table 2). The intake of bread was negatively correlated with the relative abundance of *Faecalibacterium* delta, *Lactobacillus* delta, and *Dorea* delta (Figure 4 and Table 2). The intake of potato was positively correlated with the relative abundance of *Bacteroides* delta and negatively correlated with the relative abundance of *Akkermansia* delta and *Subdoligranulum* delta (Figure 4 and Table 2).

## 4. Discussion

We analyzed the alterations in the gut microbiome in Japanese patients with T2D associated with the intervention of acarbose and its association with habitual dietary intake of the subjects to reveal the potential effects of this drug on T2D pathogenesis through the gut microbiome. 

### 4.1. Alterations of the Gut Microbial Species Associated with the Intervention of Acarbose

In the present study, the abundances of *Bifidobacterium* and *Lactobacillus* were significantly increased, whereas the abundance of *Bacteroides* was significantly decreased after the acarbose treatment (Figure 2 and Appendix A), which was consistent with previous papers [7,12,14,26,27,28]. Many members of the genus *Bifidobacterium* were reported to produce formate and acetate under carbohydrate-deprived conditions, whereas they produce lactate and acetate under carbohydrate-enriched conditions [29]. Then, specific gut microbial species convert lactate into butyrate and propionate [29]. Given that acarbose creates a relatively carbohydrate-rich environment by delivering unabsorbed sugars to the large intestine, the acarbose intervention and subsequent increase in the abundance of the *Bifidobacterium* genus can be associated with the production of short chain fatty acids (SCFA) in the gut of patients with T2D. Additionally, the abundance of genus *Eubacterium* was also significantly increased related to the acarbose intervention in this study (Figure 2 and Appendix A). *Eubacterium rectale*, which is known to be the dominant species in the gut microbiome of healthy Japanese individuals [30], is a representative source of butyrate in the gut [31,32,33]. Published data have revealed that acarbose intervention is associated with increased SCFA production in the gut of both human and murine models [28,34,35,36]. Thus, it is possible to speculate on a partial causality between acarbose treatment and T2D pathogenesis through the alteration of gut microbiome because accumulating evidence suggests that the lack of gut-derived SCFA is closely associated with the pathogenesis of T2D [2,37,38,39]. Moreover, another study revealed that acarbose administration markedly increased hydrogen production in the gut [40]. As hydrogen is a ubiquitous molecule with antioxidant property [41], acarbose-related microbial changes might influence T2D pathogenesis.

### 4.2. Associations between Habitual Dietary Nutrient Intake and Alterations of Microbial Data Associated with Acarbose Intervention

We analyzed the association between the habitual dietary nutrient intake and changes in major microbial data associated with acarbose intervention because previous papers did not touch on this issue, even though it is known that the efficacy of acarbose depends on dietary intake [42,43]. Surprisingly, clustering analysis classified the 12 major dietary nutrient items into the three rational clusters based on the pattern of the changes in the composition of the gut microbiome (Figure 3). Cluster I consisted of sucrose only, whereas cluster II included animal protein, animal fat, saturated fat, cholesterol, monounsaturated fat, polyunsaturated fat, and vegetable fat, most of which are related to fat intake. Cluster III included vegetable protein, insoluble dietary fiber, soluble dietary fiber, and carbohydrates, most of which are associated with carbohydrate intake. These findings suggest that habitual nutrient intake can affect the microbial changes related to acarbose in the gut of patients with T2D.

We also evaluated the association between the intake of various staple foods and changes in the microbial composition at the genus level accompanied by acarbose intervention. The amount of change in the major butyrate-producing genus; *Faecalibacterium* was positively correlated with the intake of rice, but negatively correlated with the intake of bread. The intake of potato was negatively correlated with the variation of two genera: *Akkermansia* and *Subdoligranulum* (Figure 4). In addition to butyrate-producing genus *Faecalibacterium*, the two genera; *Akkermansia* and *Subdoligranulum*, were reported to have a positive effect on the pathogenesis of T2D because *Akkermansia* counteracts diet-induced obesity and related disorders by enhancing the gut barrier function and moderating inflammatory responses [44] and *Subdoligranulum* is associated with the improvement of various clinical parameters regarding T2D [45]. “Washoku” is a traditional Japanese food which is famous as a healthy diet due to its low-calorie, nutritious, and well-balanced properties. Washoku was registered as a UNESCO Intangible cultural heritage in 2013. Rice is mainly contained in Washoku as a staple food rather than bread or potato. Based on the present results regarding the association between total intake of each staple food and gut microbial changes associated with acarbose intervention, acarbose has a good compatibility with Washoku, which may be helpful for the patient selection to use acarbose and dietary guidance to enhance the efficacy of this drug.

### 4.3. Limitations

There were some limitations to this study. As this was a pilot, open-label, and single-armed design, and our patient cohort was considerably small and derived from a single facility, the results should be confirmed by two-armed design, large-scale, multicenter studies. Moreover, evaluation of the fecal metabolites, including SCFA, is preferable because it appears to be informative for confirming the obtained results; however, we lacked an adequate supply of cryopreserved fecal samples. Multi-omics analysis will be required to further clarify the effects of acarbose on the pathogenesis of T2D through the gut microbiome [46]. In addition, in the present study, we collected fecal samples with a kit containing DNA stabilizer. It is known that the different isolation protocols may vary in terms of efficiency, depending on the physical and chemical matrix of the sample; and, thus, the recommended kits for the isolation of DNA from stool samples might be desirable [47]. In addition, we selected 5000 reads per sample were used for the evaluation, and therefore there is a possibly that the analysis might not be very deep; however the Good’s coverage index of the 5000 reads per sample in this study exceeded 0.98, indicating a high coverage degree which was a sufficient reads number for this fecal microbiome analysis. We plan to change the method to the recommended one in the future studies. Further, even though we did not instruct the patients to alter their lifestyles and dietary contents, we cannot confirm it in the present study. Another limitation was that we could not eliminate the effect of other medications prescribed to the patients. Certainly, some antidiabetic drugs, such as metformin, were reported to affect the composition of the gut microbiome. In the present study, metformin was used in 14 patients, which may have affected the obtained results, including the greater abundance of *Bifidobacterium* and *Lactobacillus*. Nevertheless, we suppose that their effects are quite limited, because we just focused on the alteration of gut microbiome and no previous research reported the synergistic effect of acarbose and another drug on the composition of the gut microbiome.

## 5. Conclusions

We revealed the characteristics of gut microbiome altered by acarbose intervention and also identified the possible relationship between habitual dietary intake and gut microbial changes in Japanese patients with T2D.

## Figures and Tables

**Figure 1 nutrients-13-02107-f001:**
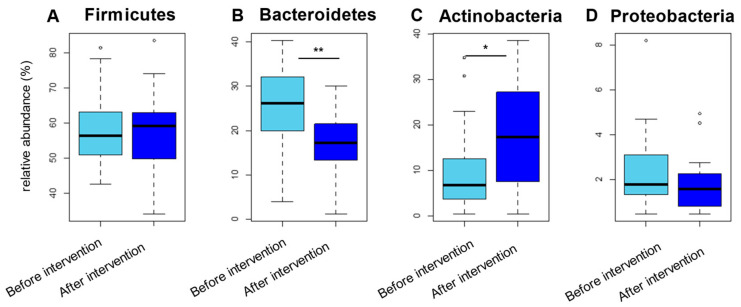
Alterations in the microbiome at the phylum-level associated with the intervention of acarbose. (**A**), Firmicute; (**B**), Bacteroidetes; (**C**), Actinobacteria; (**D**), Proteobacteria. Comparison of the microbial abundance at the phylum-level between the two groups (before and after the intervention of acarbose) based on 16S data. Top four abundant phyla are represented. Each box plot represents median, interquartile range, minimum, and maximum values; * *p* < 0.05 and ** *p* < 0.01, based on the paired *t*-test. (**A**), represents phylum Firmicutes; (**B**), represents phylum Bacteroides; (**C**), represents phylum Actinobacteria; and (**D**), represents phylum Proteobacteria.

**Figure 2 nutrients-13-02107-f002:**
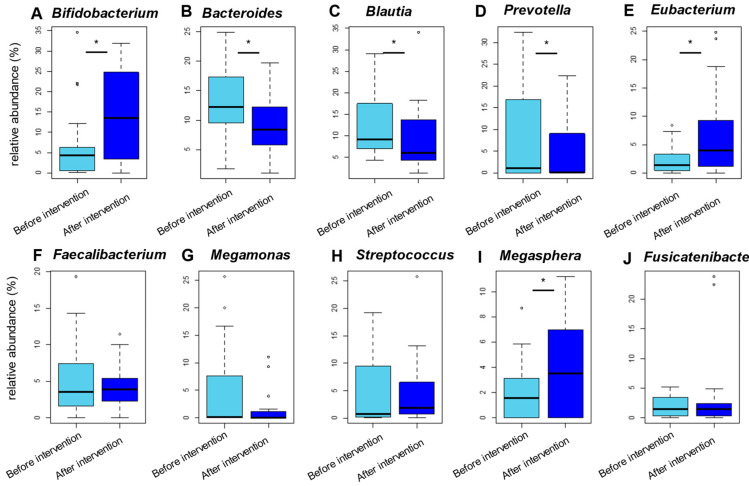
Alterations in the microbiome at the genus-level associated with the intervention of acarbose. Comparison of the abundance at the genus-level before and after the intervention of acarbose based on 16S data. (**A**), Bifidobacterium; (**B**), Bacteroides; (**C**), Blautia; (**D**), Prevotella; (**E**), Eubacterium; (**F**), Faecalibacterium; (**G**), Megamonas; (**H**), Streptococcus; (**I**), Megasphera; (**J**), Fusicatenibacte.Top ten abundant genera are represented. Each box plot shows median, interquartile range, minimum, and maximum values; * *p* < 0.05 based on the paired *t*-test. (**A**), represents genus *Bifidobacterium*; (**B**), represents genus *Bacteroides*; (**C**), represents genus *Blautia*; (**D**), represents genus *Prevotella*; (**E**), represents genus *Eubacterium*; (**F**), represents genus *Faecallbacterium*; (**G**), represents genus *Megamonas*; (**H**), represents genus *Streptococcus*; (**I**), represents genus *Megasphera*; and (**J**), represents genus *Fusicatenibacte*.

**Figure 3 nutrients-13-02107-f003:**
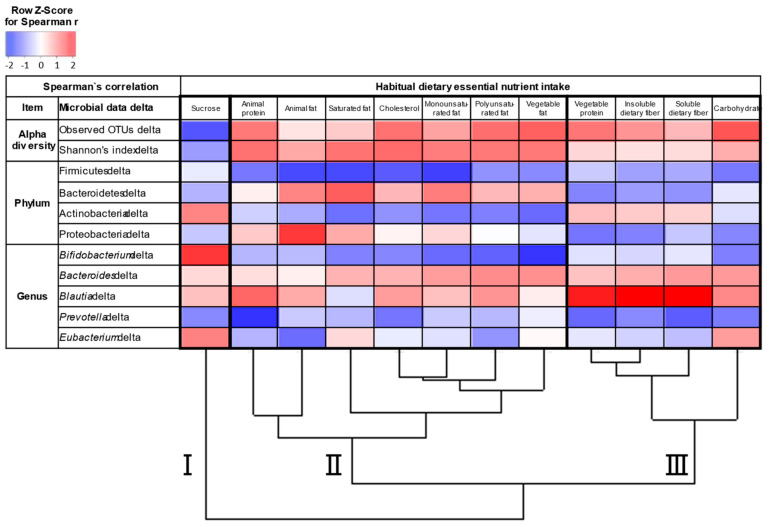
Association between habitual dietary intake and changes in microbial data. Spearman’s correlation of habitual dietary nutrient intake with the observed OTU number delta, Shannon index delta, and average relative abundance delta of the top four phyla and top five genera using a heatmap. The 12 dietary nutrient items were used for hierarchical clustering. The z-score based on the Spearman’s r between changes in the composition of the gut microbiome and habitual dietary nutrient intake is depicted from the lowest (blue) to the highest (red) according to the scale shown at the top. X delta = X after intervention − X before intervention.

**Figure 4 nutrients-13-02107-f004:**
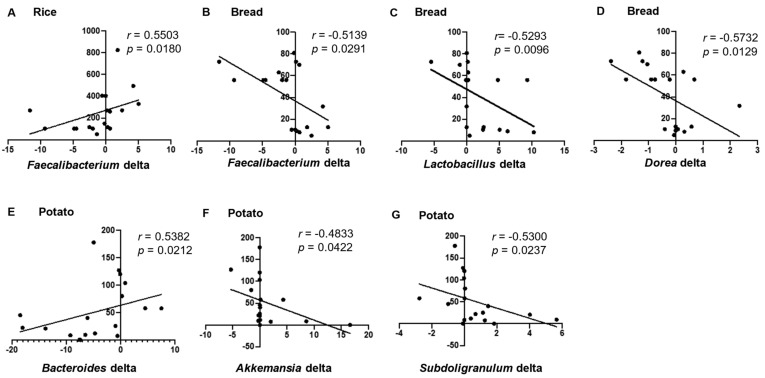
Correlation between the intake of various staple foods and changes in microbial data. Spearman’s correlation between the relative abundance delta of microbial data and the intake of various staple foods with statistical significance. X delta = X after intervention − X before intervention. (**A**), represents correlation between rice intake and *Faecallbacterium* delta; (**B**), represents correlation between bread intake and *Faecallbacterium* delta; (**C**), represents correlation between bread intake and *Lactobacillus* delta; (**D**), represents correlation between bread intake and *Dorea* delta; (**E**), represents correlation between potato intake and *Bacteroides* delta; (**F**), represents correlation between potato intake and *Akkemanisa* delta; and (**G**), represents correlation between potato intake and *Subdoligranuium* delta.

**Table 1 nutrients-13-02107-t001:** Demographics and characteristics of the 18 patients with type 2 diabetes.

	T2D before Intervention (n = 18)	T2D after Intervention (n = 18)	*p*-Value
Age, year	66.5 (48–75)	66.5 (48–75)	-
Sex (female:male)	9:9	9:9	-
Dose (150 mg/day:300 mg/day)	13:5	13:5	-
BMI, kg/m^2^	22.8 (19.0–35.9)	23.0 (19–35.9)	0.20
Systolic BP, mmHg	125.0 (86–169)	120.0 (86–154)	0.19
Diastolic BP, mmHg	73.0 (54–95)	73.0 (54–86)	0.11
Laboratory examination			
HbA1c, %	7.4 (6.6–9.9)	7.3 (6.4–9.6)	0.07
Cre, mg/dL	0.66 (0.49–1.04)	0.65 (0.50–1.03)	0.58
eGFR, mL/min/1.73 m^2^	76.2 (58.2–113.8)	73.9 (58.9–136.3)	0.91
AST, IU/L	21.0 (13–42)	27.0 (13–45)	0.001
ALT, IU/L	20.0 (10–59)	27.0 (10–74)	0.01
GTP, IU/L	27.0 (10–94)	23.0 (10–150)	0.21
T-cho, mg/dL	211.0 (155–284)	229.0 (106–288)	0.84
LDL-cho, mg/dL	132.0 (78–169)	145.5 (86–191)	0.04
HDL-cho, mg/dL	52.0 (35–113)	55.0 (37–101)	0.75
TG, mg/dL	94.5 (60–301)	104.0 (45–257)	0.27
WBC, /μL	5700 (3500–)	6250 (3300–9100)	0.67

Data are represented as median (range). Abbreviations: T2D = type 2 diabetes; BMI = body mass index; BP = blood pressure; HbA1c = hemoglobin A1c; Cre = creatine; eGFR = estimated glomerular filtration rate; AST = aspartate transaminase; ALT = alanine aminotransferase; GTP = glutamyltransferase; T-cho = total cholesterol; LDL-cho = LDL-cholesterol; HDL-cho = HDL-cholesterol; TG = triglyceride; WBC = white blood cells.

**Table 2 nutrients-13-02107-t002:** Association between the intake of various staple foods and changes in microbial data.

Spearman’s Correlation	Habitual Dietary Intake			
	Rice	Bread	Noodle	Potato
*Bifidobacterium* delta	−0.15	−0.04	0.02	−0.11
*Bacteroides* delta	0.01	0.13	−0.18	**0.53** *
*Blautia* delta	0.01	0.41	0.21	0.34
*Prevotella* delta	0.16	−0.08	−0.35	0.17
*Eubacterium* delta	0.12	−0.25	−0.11	0.02
*Faecalibacterium* delta	**0.55 ***	−**0.51** *	0.19	−0.36
*Megamonas* delta	−0.01	−0.09	−0.05	−0.08
*Streptococcus* delta	−0.19	0.05	−0.25	0.12
*Megasphaera* delta	−0.01	0.27	−0.05	0.22
*Fusicatenibacter* delta	−0.06	−0.40	−0.19	0.04
*Collinsella* delta	0.06	−0.04	0.36	−0.28
*Clostridium* delta	0.09	0.20	−0.12	0.22
*Akkermansia* delta	0.21	0.03	0.21	−**0.48** *
*Lactobacillus* delta	−0.24	−**0.59** *	−0.32	−0.15
*Subdoligranulum* delta	−0.32	0.13	0.02	−**0.53** *
*Ruminococcus* delta	−0.09	0.45	0.38	−0.06
*Dorea* delta	0.17	−**0.57** *	−0.25	−0.06
*Parabacteroides* delta	−0.09	0.29	0.15	−0.07
*Alistipes* delta	−0.02	0.38	−0.01	−0.29
*Acidaminococcus* delta	0.13	0.05	−0.11	0.03
*Veillonella* delta	−0.15	−0.17	−0.05	−0.08
*Phascolarctobacterium* delta	0.04	0.03	−0.40	0.06
*Lachnoclostridium* delta	−0.46	0.37	−0.16	0.39
*Anaerostipes* delta	−0.05	0.28	0.07	0.21
*Escherichia* delta	−0.05	0.45	0.22	0.15
*Holdemanella* delta	0.24	−0.09	−0.24	0.28
*Catenibacterium* delta	0.09	0.15	0.25	−0.14
*Sutterella* delta	−0.07	−0.44	−0.46	0.04
*Dialister* delta	−0.08	0.09	0.09	0.23
*Roseburia* delta	0.37	−0.10	−0.08	0.21
*Oscillibacter* delta	0.12	0.23	0.40	−0.04
*Fusobacterium* delta	0.05	−0.15	−0.25	0.19

Spearman’s correlation of the intake of various staple foods with average relative abundance delta of the top 32 genera; * *p* < 0.05 based on the Spearman’s rank correlation coefficient. X delta = X after intervention − X before intervention.

## Data Availability

The data that support the findings of this study are available from the corresponding author upon reasonable request.

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
