# Peer review of "Habitual Dietary Intake Affects the Altered Pattern of Gut Microbiome by Acarbose in Patients with Type 2 Diabetes"

_nutrients, 2021, doi:10.3390/nu13062107_

Round 1

Reviewer 1 Report

The methodology does not indicate when the samples were collected, they should indicate the days before and after the intervention.

Author Response

  • Response to Reviewer #1

The methodology does not indicate when the samples were collected, they should indicate the days before and after the intervention.

# Response:

We appreciate the reviewer’s helpful advice. The samples were collected a day before and four weeks after the start of intervention. Accordingly, we have modified the description, as follows.

Original; Page 2, Line 75-76: Fecal and blood samples were gathered before and four weeks after the intervention.

Corrected; Page 2, Line 75-76: Fecal and blood samples were gathered a day before and four weeks after the start of intervention.

Reviewer 2 Report

Manuscript: nutrients-1235279

Article type: full-length article

Title: Habitual Dietary Intake Affects the Altered Pattern of Gut Microbiome by Acarbose in Patients with Type 2 Diabetes

Authors: Fumie Takewaki, Hanako Nakajima, Daiki Takewaki, Yoshitaka Hashimoto *, Saori Majima, Hiroshi Okada, Takafumi Senmaru, Emi Ushigome, Masahide Hamaguchi, Masahiro Yamazaki, Yoshiki Tanaka, Shunji Nakajima, Hiroshi Ohno, Michiaki Fukui

Journal: Nutrients

General comments:

The submitted article describes the microbiota composition of Japanese patients with type 2 diabetes treated by acarbose and evaluated the possible association with habitual dietary intake. The manuscript is very well written without any typos and spelling mistakes. The figures and tables are properly and carefully prepared. Obtained results are clear and there are no major shortcomings. However, there are some issues to be discussed, mainly concerning microbiome analysis.

Major comments:

1) Authors isolated DNA from fecal samples according to their procedure. However, it is known that the different isolation protocols may vary in terms of efficiency, depending on the physical and chemical matrix of the sample. Consequently, the analysis of microbial community diversity or the quantification of specific genes is influenced by the DNA extraction method, reflected in its efficiency. DNA extraction method strongly affects the results of studies on the diversity and structure of soil microbial community. Choosing the most appropriate method for DNA extraction is very important. The greatest diversity of microorganisms in the sample is desirable.

Therefore, the recommended kits for the isolation of DNA from stool samples should be used (eg. by Illumina).

2) The main complaint concerns the analysis of obtained microbiota data. Authors say that 50,000 raw reads were randomly gathered from the sequence files for each sample. (l. 166). But then, only 5,000 reads per sample were used for the evaluation of the alpha and beta diversities (l. 206). Why? That's terribly little. With amplicon sequencing normally you should receive, for instance, 30,000 – 150,000 read pairs per sample. To get good coverage and deep enough, you need to have about 10,000 reads per sample. However, you will be able to find new species/genera even if you sequence up to 100,000 reads per sample. The rare species could be very rare.

With 5,000 reads per sample used in this analysis, authors lose a lot of rare OTUs/species/genera. Therefore, the analysis is not very deep and has bad coverage. I understand that 5,000 reads are probably the lowest number in a particular sample. But it is always better to omit some low samples than to decrease the efficiency of the whole analysis.

As a consequence, the authors found only 32 genera with an abundance >0.1%, which is a really low number. (l.277). And another consequence is an insufficiently deep analysis, which is reflected in the possibility of determining diversity only at the level of bacterial genera and not strains. It is known that even different OTUs belonging to the same genera have different activities and effects. The analysis should be performed at the OTUs level rather than the genera level.

3) The study group is relatively small for the plausibility of the results, but the authors themselves mentioned this as a limitation of this study.

Minor comments:

  • 229, 232, 281 – Instead of putting p values in the text, I would prefer to include text tables that would make the text clearer.
  • 303-304 – The sentence is incomplete. The abundances were increased or decreased…where? Probably after the treatment of patients with acarbose….

Author Response

● Response to Reviewer #2

Major comments:

  1. Authors isolated DNA from fecal samples according to their procedure. However, it is known that the different isolation protocols may vary in terms of efficiency, depending on the physical and chemical matrix of the sample. Consequently, the analysis of microbial community diversity or the quantification of specific genes is influenced by the DNA extraction method, reflected in its efficiency. DNA extraction method strongly affects the results of studies on the diversity and structure of soil microbial community. Choosing the most appropriate method for DNA extraction is very important. The greatest diversity of microorganisms in the sample is desirable. Therefore, the recommended kits for the isolation of DNA from stool samples should be used (eg. by Illumina).

# Response:

I really appreciate the reviewer’s appropriate comment regarding DNA extraction. In the present study, we collected fecal samples with a kit containing DNA stabilizer, and all collected samples were used for DNA extraction. Therefore, there are no residual fecal samples. Last year, a paper on comparison of DNA extraction methods was published [reference A]. Therefore, we plan to change the method to the recommended one in the future studies.

Reference A; Tourlousse DM, Narita K, Miura T, Sakamoto M, Ohashi A, Shiina K, et al. Validation and standardization of DNA extraction and library construction methods for metagenomics-based human fecal microbiome measurements. Microbiome 2021; 9: 95.

According to the reviewer’s comment, we have added the following sentences in the Discussion section.

Original; none

Corrected; Page 11, Line 363-368: In addition, in the present study, we collected fecal samples with a kit containing DNA stabilizer. It is known that the different isolation protocols may vary in terms of efficiency, depending on the physical and chemical matrix of the sample; and thus, the recommended kits for the isolation of DNA from stool samples might be desirable [47]. We plan to change the method to the recommended one in the future studies.

Reference 47. Tourlousse DM, Narita K, Miura T, Sakamoto M, Ohashi A, Shiina K, et al. Validation and standardization of DNA extraction and library construction methods for metagenomics-based human fecal microbiome measurements. Microbiome 2021; 9: 95.

2-1) The main complaint concerns the analysis of obtained microbiota data. Authors say that 50,000 raw reads were randomly gathered from the sequence files for each sample. (l. 166). But then, only 5,000 reads per sample were used for the evaluation of the alpha and beta diversities (l. 206). Why? That's terribly little. With amplicon sequencing normally you should receive, for instance, 30,000 – 150,000 read pairs per sample. To get good coverage and deep enough, you need to have about 10,000 reads per sample. However, you will be able to find new species/genera even if you sequence up to 100,000 reads per sample. The rare species could be very rare. With 5,000 reads per sample used in this analysis, authors lose a lot of rare OTUs/species/genera. Therefore, the analysis is not very deep and has bad coverage. I understand that 5,000 reads are probably the lowest number in a particular sample. But it is always better to omit some low samples than to decrease the efficiency of the whole analysis.

# Response:

I really appreciate the reviewer’s important comments. We have three reasons why we chose 5,000 reads per sample for the microbiome analysis. As the reviewer pointed out, the first reason is that the lowest number in a particular sample was about 5,000 reads. In order to align the sequence depth in all samples, we chose this number. The second reason is that 16S-based analysis using too many sequencing reads causes the overestimation of the species richness in the clustering due to intrinsic sequencing error. We cannot avoid the influence of sequencing error in evaluating very rare OTUs/species/genera in this kind of 16S rDNA analysis. The third reason is that the Good's coverage index [reference B] of the 5,000 reads per sample in this study exceeded 0.98, indicating a high coverage degree which was sufficient reads number for this fecal microbiome analysis.

Reference B; Kim, S. W. et al. Robustness of gut microbiota of healthy adults in response to probiotic intervention revealed by high-throughput pyrosequencing. DNA Res. 20, 241–253 (2013).

According to the reviewer’s comment, we have added the following sentences in the method section.

Original; none

Corrected; Page 4, Line 170-175:

We randomly selected 5,000 reads per sample, and analyzed them to minimize the overestimation of the species richness in the clustering due to intrinsic sequencing error. The Good's coverage index of the 5,000 reads per sample in this study exceeded 0.98, indicating a high coverage degree which was sufficient reads number for this fecal microbiome analysis [24].

Reference 24; Kim SW, Suda W, Kim S, Oshima K, Fukuda S, Ohno H, et al. Robustness of gut microbiota of healthy adults in response to probiotic intervention revealed by high-throughput pyrosequencing. DNA Res 2013; 20: 241–253.

2-2) As a consequence, the authors found only 32 genera with an abundance >0.1%, which is a really low number. (l.277). And another consequence is an insufficiently deep analysis, which is reflected in the possibility of determining diversity only at the level of bacterial genera and not strains. It is known that even different OTUs belonging to the same genera have different activities and effects. The analysis should be performed at the OTUs level rather than the genera level.

# Response:

I really appreciate the reviewer’s appropriate comment. As this is an exploratory research, we first evaluated the microbiota using 16S rDNA. The 16S rDNA analysis targeted 460 base pairs, which contains only two variable regions. Therefore, it is not suitable for the analysis at the OTU/species level. We would like to consider the whole metagenomic sequencing analysis to secure the species level resolution in the future studies.

3.The study group is relatively small for the plausibility of the results, but the authors themselves mentioned this as a limitation of this study.

# Response:

We appreciate the reviewer’s helpful comment and completely agree with you. Our patient cohort was relatively small, the results should be confirmed by large scale, multicenter studies.

Minor comments:

229, 232, 281 – Instead of putting p values in the text, I would prefer to include text tables that would make the text clearer.

# Response:

We agreed with the reviewer’s opinion and have deleted p values in the text.

303-304 – The sentence is incomplete. The abundances were increased or decreased…where? Probably after the treatment of patients with acarbose….

# Response:

We thank the reviewers for this helpful advice. We have modified the descriptions, as follows.

Original; Page 8, Line 303-304: the abundances of Bifidobacterium and Lactobacillus were significantly increased, whereas the abundance of Bacteroides was significantly decreased

Corrected; Page 7-8, Line 300-302: the abundances of Bifidobacterium and Lactobacillus were significantly increased, whereas the abundance of Bacteroides was significantly decreased after the acarbose treatment

Reviewer 3 Report

A WELL WRITTEN AND INTERESTING PAPER REGARDING THE ACARBOSE-ASSOCIATED GUT MICROBIOME CHANGES IN TYPE 2 DM. AS USE OF PROBIOTICS AND METABIOTICS IS INREASING  WRITTERS MAY CONSIDER A COMMENT REGARDING THESE PRODUCTS AND ITS POTENTIAL BENEFITS BASED ON THE RESULTS OF THE PRESENT PAPER. 

Author Response

  • Response to Reviewer #3

A WELL WRITTEN AND INTERESTING PAPER REGARDING THE ACARBOSE-ASSOCIATED GUT MICROBIOME CHANGES IN TYPE 2 DM. AS USE OF PROBIOTICS AND METABIOTICS IS INREASING WRITTERS MAY CONSIDER A COMMENT REGARDING THESE PRODUCTS AND ITS POTENTIAL BENEFITS BASED ON THE RESULTS OF THE PRESENT PAPER. 

# Response:

I really appreciate the reviewer’s precious comment. We will use this comment as reference for the future research.